# MILES: Making Imitation Learning Easy with Self-Supervision

**Georgios Papagiannis and Edward Johns**
The Robot Learning Lab
Imperial College London, UK
`g.papagiannis21@imperial.ac.uk`

**Abstract:** Data collection in imitation learning often requires significant, laborious human supervision, such as numerous demonstrations, and/or frequent environment resets for methods that incorporate reinforcement learning. In this work, we propose an alternative approach, MILES: a fully autonomous, self-supervised data collection paradigm, and we show that this enables efficient policy learning from just a single demonstration and a single environment reset. MILES autonomously learns a policy for returning to and then following the single demonstration, whilst being self-guided during data collection, eliminating the need for additional human interventions. We evaluated MILES across several real-world tasks, including tasks that require precise contact-rich manipulation such as locking a lock with a key. We found that, under the constraints of a single demonstration and no repeated environment resetting, MILES significantly outperforms state-of-the-art alternatives like imitation learning methods that leverage reinforcement learning. Videos of our experiments and code can be found on our webpage: www.robot-learning.uk/miles.

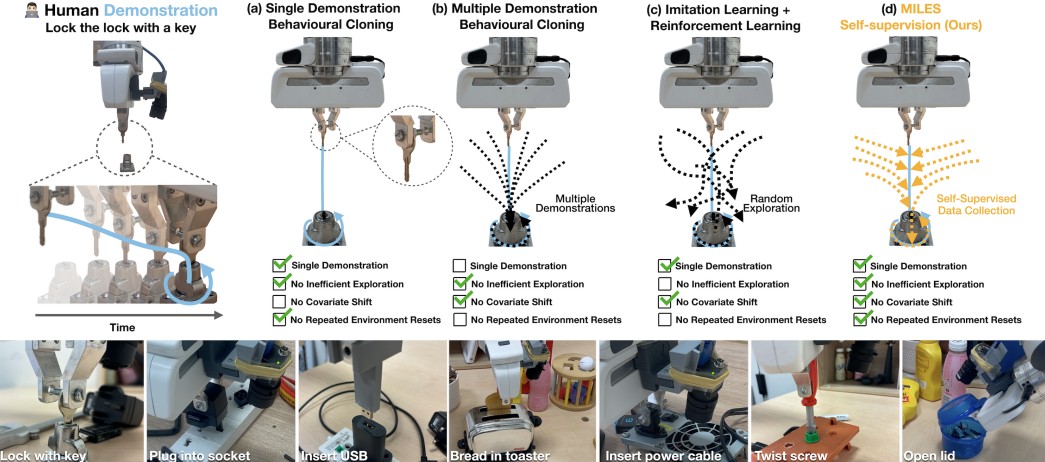

Figure 1: (a) Behavioural cloning from a single demonstration fails to generalize to states outside the demonstration, due to covariate shift. (b) Providing multiple demonstrations addresses this, but requires significant human effort. (c) While incorporating reinforcement learning addresses the issue of covariate shift and the need for multiple demonstrations, it requires frequent environment resetting and is highly inefficient due to random exploration. (d) In MILES, we propose a new self-supervised paradigm that overcomes these issues and can learn a range of complex tasks from a single demonstration and no additional human effort, by collecting augmentation trajectories that guide the robot back to the demonstration.

## 1 Introduction

Imitation learning is frequently described as a convenient way to teach robots new skills. But is this true in practice? Behavioral cloning (BC) methods leverage supervised learning to train robust policies, but doing so typically requires hundreds or thousands of demonstrations per task [1, 2] to collect a sufficiently diverse training dataset. Imitation learning methods that leverage Reinforcement learning (RL) offer a solution to this as policies can be learned autonomously through random exploration and reward functions can be inferred from a few demonstrations [3, 4]. However, unlike supervised learning methods, policy learning with RL can be unstable [5], and random exploration

8th Conference on Robot Learning (CoRL 2024), Munich, Germany.

makes data collection inefficient. Besides, RL typically requires repeated environment resetting, and so in practice, it is often equally or even more laborious than simply providing numerous demonstrations [6]. Instead, learning from a single demonstration appears to be the most convenient form of imitation learning due to its effortlessness, but policies learned this way suffer from covariate shift [7]. As such, imitation learning today is not as easy as we would like it to be: significant human supervision is still required for data collection either via demonstrations, environment resetting, or both.

Motivated by this, we propose MILES, a framework that makes imitation learning easy by enabling humans to teach robots tasks effortlessly with just a *single* demonstration while requiring no prior task knowledge, and only a single environment reset. MILES learns robot skills by collecting *augmentation trajectories* in a self-supervised manner, which demonstrate to the robot how to return to, and then follow the single human demonstration, as shown in Figure 1. MILES leverages these trajectories to train a policy using BC, consequently inheriting the power and ease of supervised learning. However, unlike common BC methods, the self-supervised data collection replaces the need to collect multiple demonstrations, and compared to BC from a single demonstration, MILES does not suffer from covariate shift as its data densely covers the space around the demonstration. Similarly to RL-based imitation learning methods, MILES collects data autonomously, but instead of being guided by random exploration, MILES' data collection is highly efficient as it benefits from self-labeled data that directly guides the robot back to the demonstration. Additionally, exploration can cause disturbances to the environment and thus requires repeated environment resetting. However, as we later explain, MILES' self-supervised data collection procedure strategically shapes policy learning to be independent of such laborious environment resets. Consequently, MILES aims to incorporate the benefits of training BC policies with supervised learning on large demonstration datasets with the benefits of autonomous data collection similar to RL methods in the setting where only a single demonstration is available.

Through our real-world experiments, we find that when only a single human demonstration is available, without additional human effort such as repeated environment resetting, self-supervised data collection outperforms recent alternatives in imitation learning that leverage RL and replay-based imitation learning. MILES can learn a suite of versatile skills, shown in Figure 1, ranging from those involving interactions with movable, articulated objects such as opening the lid of a box, to those involving precise, contact-rich interactions with objects rigidly mounted to a surface, such as inserting and twisting a key in a lock. Each of these is learned from a single demonstration, no further human input, and around 30 minutes of self-supervised data collection.

## 2   Related Work

As follows, we ground our work relative to methods that can learn manipulation skills from a single demonstration, unlike most approaches that require large demonstration datasets [1, 8, 9].

**Imitation learning from prior knowledge.** An effective way to compensate for the lack of large demonstration datasets is to leverage prior task knowledge such as access to ground truth object poses [10, 11] or by meta-learning policies by first pretraining on large demonstration datasets [12, 13]. However, precise knowledge of the objects' poses is hard to obtain in practice and meta-learning methods are often limited to tasks similar to the ones seen in the demonstrations. Instead, MILES can learn a new task from just a single demonstration without any prior object or task knowledge.

**Imitation learning via Reinforcement learning (RL).** RL-based imitation learning methods from a single demonstration learn to follow that demonstration by minimizing a similarity metric between the trajectories of the learned policy and the demonstration [3, 4, 14, 15]. Other RL methods that learn from demonstrations infer rewards through alternative means, like goal images [16]. Though effective, these methods are often inefficient as they rely on random exploration and repeated environment resets which require significant human effort. Instead, our self-supervised data collection makes MILES highly efficient and eliminates the need for repeated environment resetting.

**Imitation learning via pose estimation and demonstration replay.** Replay-based imitation learning methods first estimate and move the robot to a similar pose relative to the objects of interest as in the demonstration and then replay the demonstrated robot actions [17, 18, 19, 20, 21]. While these methods are the most efficient in terms of human time, small errors in pose estimation cause errors to compound during demonstration replay, leading to task failures [2]. And even under the assumption of perfect pose estimation, potential environment collisions may prevent the robot from reaching the desired pose or may perturb the objects such that replaying the demonstration fails to complete

the task. Instead, MILES' self-supervised data collection procedure retains the human-time efficiency of pose estimation methods, while learning to avoid unnecessary collisions, and minimizing or completely eliminating open-loop replay errors depending on the task.

**Imitation learning by demonstration augmentation.** Demonstration augmentation approaches like DAgger [7] and DART [22] mitigate covariate shift by relying on laborious interactive expert queries to expand the known state distribution of a policy. And methods that do not require an interactive expert still rely on multiple demonstrations or task-specific optimizations [23, 24, 25, 26, 27] which limit their practical application. Instead, MILES is a fully autonomous method that uses self-supervision to augment a single demonstration and can learn a wide range of diverse tasks.

## 3 MILES: Making Imitation Learning Easy with Self-Supervision

As follows we describe MILES, a framework that makes imitation learning easy by leveraging a single human demonstration as guidance to collect self-supervised data demonstrating to the robot how to return to, and then follow the demonstration. By training a policy with behavioral cloning on that data, MILES learns to perform a task from a range of initial states and object poses.

### 3.1 Preliminaries

**Assumptions.** Our setup assumes access to a wrist camera that is rigidly mounted to the robot's end-effector (EE) and (optionally) a sensor that measures external forces and torques. We follow prior work [19, 2, 17] and assume that each task is object-centric, such that only the task-relevant object is in camera view during data collection and the demonstration can be expressed relative to a single object, where combining several such tasks results in a multi-stage task. Additionally, as we are interested in dealing with all types of tasks, including those that require contact-rich manipulation, we control our robot using an impedance controller.

**Single Demonstration.** For each task, a human provides a *single* demonstration $\zeta :=$ $\{(w_n^\zeta, o_n^\zeta, a_n^\zeta)\}_{n=1}^N$ comprising a sequence of $N$ waypoints $w_n^\zeta$, observations $o_n^\zeta$, and actions $a_n^\zeta$, as shown in Figure 2 (1). A waypoint $w_n^\zeta$ corresponds to the EE's 6-DoF pose at timestep $n$ captured via proprioception. An observation $o_n^\zeta$ consists of an RGB image captured from the wrist camera and a force-torque measurement. We refer to $(w_n^\zeta, o_n^\zeta)$ as the state of the environment at timestep $n$. An action $a_n^\zeta$ contains the gripper's state and the 6-DoF pose tracked by the impedance controller at timestep $n$, expressed relative to the EE's pose at timestep $n-1$. After providing $\zeta$ the human resets the environment *only once*, such that if the actions in $\zeta$ are executed, the robot would successfully perform the task; a trivial process that requires a few seconds of human time.

### 3.2 Self-Supervised Data Collection

**Augmentation Trajectories.** Given a single demonstration, MILES collects a dataset of augmentation trajectories $\mathcal{D} := \{\tau_k\}$, where $1 \le k \le N$ and each $\tau_k := \{(w_m^{\tau_k}, o_m^{\tau_k}, a_m^{\tau_k})\}_{m=1}^M$ is a robot trajectory whose final, $M_{\text{th}}$ state corresponds to a $k_{\text{th}}$ state in the demonstration, i.e., $(w_M^{\tau_k}, o_M^{\tau_k}) = (w_k^\zeta, o_k^\zeta)$. That is, each augmentation trajectory guides the robot to some $k_{\text{th}}$ state in the demonstration from any state $(w_m^{\tau_k}, o_m^{\tau_k}) \in \tau_k$. We can *fuse* each augmentation trajectory with the demonstration segment following each $k_{\text{th}}$ state, $\{(w_n^\zeta, o_n^\zeta, a_n^\zeta)\}_{n=k}^N \subseteq \zeta$, to create a *new* demonstration that demonstrates to the robot how to **return to** and then **follow** the human demonstration as shown in Figure 3. By collecting augmentation trajectories that densely cover the state space near the demonstration we can create a dataset of new demonstrations which we can leverage to train a policy using standard BC methods. But how do we create such a dataset of augmentation trajectories automatically?

**Data Collection.** We achieve this by collecting data in the simplest possible way. An overview of our data collection procedure is shown in Figure 2. To generate a $\tau_k$, from a demonstration waypoint $w_k^\zeta$, we first move the robot to some random pose near the demonstration. The robot then attempts to return back to $w_k^\zeta$ in a straight line, while recording RGB and force-feedback observations $\{o_m^{\tau_k}\}_{m=1}^M$, and actions $\{a_m^{\tau_k}\}_{m=1}^M$ automatically generated by computing the EE's relative movement between consecutive timesteps, as shown in Figure 2 (3) (gripper actions are copied directly from the demonstrated action). In practice, the actual trajectory $\tau_k$ may not be a straight line, as collisions with external objects can yield a more complex, *curved*

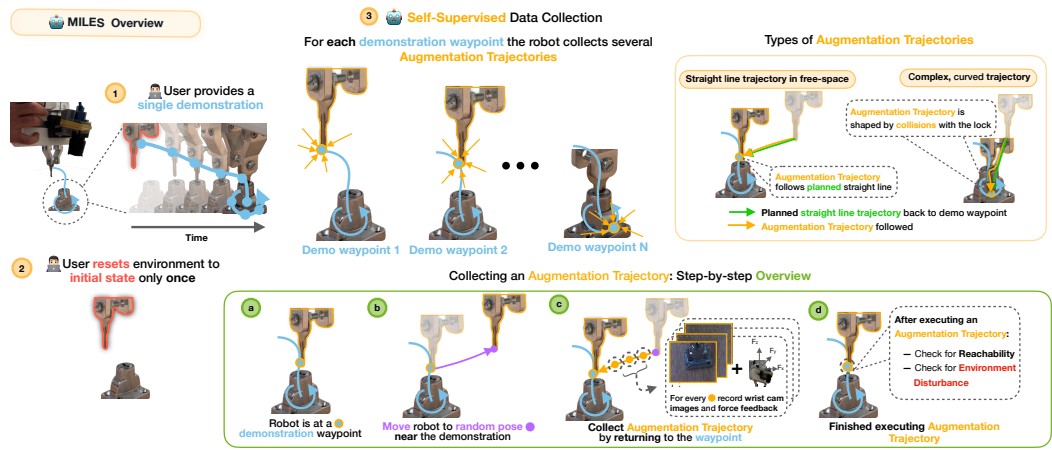

Figure 2: **MILES Overview:** (1) First, the user provides a single demonstration and (2) resets the environment only once. (3) Then, the robot (autonomously) collects self-supervised data. Several augmentation trajectories are collected for each demonstration waypoint until an environment disturbance is detected or sufficient data is collected for all waypoints. Each augmentation trajectory is either a straight line, if the motion occurs in free space, or a more complex, curved path as the augmentation trajectory can be reshaped by collisions with the environment (e.g., with the lock as shown above). (3) (a-b) To collect an augmentation trajectory, the robot first moves from a demonstration waypoint to a random pose. (c) Then, it attempts to return back to the waypoint while recording RGB images and force-torque feedback . (d) After completing the trajectory, we check whether the achieved state meets the conditions of *reachability* and *environment disturbance*.

*path* due to the robot's compliance; an example of this happening for a locking a lock task is shown in Figure 2 (3). This is particularly useful for contact-rich tasks, as these trajectories contain information on overcoming potential collisions or large friction areas by regulating force when in contact with an object. After executing the potential augmentation trajectory we check that it is valid and that it can be fused with the demonstration by evaluating two key conditions, that of **reachability** and **environment disturbance** which we describe in section 3.3. If both conditions are met we store that augmentation trajectory, otherwise, it is discarded.

We repeat this process several times for each demonstration state, starting from the first waypoint in the demonstration $w_1^\zeta$ and gradually progress through the demonstration, as shown in Figure 2 (3), until: (1) an environment disturbance is detected, in which case we stop the data collection and store the demonstration timestep where the disturbance occurred, denoted $R$, and the actions $\zeta_{remaining} := \{a_n^\zeta\}_{n=R}^N$ for the remaining demonstration states for which no data is collected; or (2) we have collected a prespecified number of $Z$ augmentation trajectories for each of the $N$ demonstration states, in which case $R = N$.

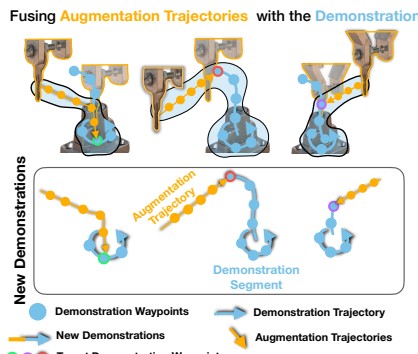

Figure 3: After finishing the data collection, each augmentation trajectory is fused with the demonstration segment following the demonstration waypoint it returns to, to create a dataset of new demonstration trajectories.

At the end of the data collection process, every augmentation trajectory is fused with $\zeta$ to form a new demonstration $\zeta_k = \{(o_m^{\tau_k}, a_m^{\tau_k})\}_{m=1}^M \cup \{(o_n^\zeta, a_n^\zeta)\}_{n=k}^R$ as shown in Figure 3. And as a result, we obtain a dataset $\mathcal{D}_{new} = \{\zeta, \zeta_1^1, \zeta_1^2, ..., \zeta_1^Z, ..., \zeta_R^{Z-1}, \zeta_R^Z\}$. Every trajectory in $\mathcal{D}_{new}$ corresponds to a *new* demonstration, automatically created, that solves the task up to the $R_{th}$ state in the demonstration. Further details concerning a practical implementation of our data collection procedure and pseudocode can be found in our supplementary material section A.1.

### 3.3 Validity Conditions for Augmentation Trajectories

As follows, we introduce two conditions that determine whether an augmentation trajectory can be fused with the human demonstration. Consider an augmentation trajectory $\tau_k$ aimed at returning the robot to the $k_{th}$ demonstration state, $(w_k^\zeta, o_k^\zeta)$:

(1) **Condition 1, Reachability:** After executing the augmentation trajectory, the EE's pose must equal the pose of the demonstration waypoint $w_k^\zeta$. This equality can be verified trivially using

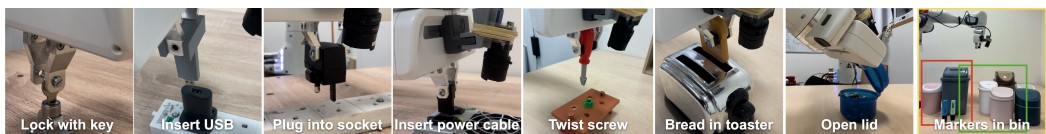

Figure 4: The tasks used in our experiments. The "Markers in Bin" is used to evaluate MILES' ability to generalize (the bins marked green denote the training set, while the red denote the test set).

proprioception. In many scenarios, the environment's dynamics (e.g. collisions) or inevitable systematic inaccuracies in a robot's controller may prevent it from reaching its target waypoint $w_k^\zeta$. Thus, if $w_M^{\tau_k} \neq w_k^\zeta$, the augmentation trajectory cannot return to demonstration state $k$, rendering the augmentation trajectory invalid.

(2) **Condition 2, Environment Disturbance:** While collecting $\tau_k$, the robot may disturb the environment, resulting in a final observation $o_M^{\tau_k}$ that no longer matches that of the demonstration (even if $w_k^\zeta$ is reached). For instance, during data collection if the robot's gripper pushes an object to a different pose than it had at timestep $k$ of the demonstration, the final observation in the augmentation trajectory will differ from the demonstration's $k_{th}$ observation. Therefore, if $o_M^{\tau_k} \neq o_k^\zeta$, the augmentation trajectory cannot be combined with the human demonstration to create a new, valid demonstration. To detect such disturbances, we compare the cosine similarity of the DINO features [28, 29] of the RGB image $I_k^\zeta$ from the demonstration's observation $o_k^\zeta$ and the image $I_M^{\tau_k}$ after executing the augmentation trajectory. If the similarity falls below a threshold $\theta$, we assume the environment has been disturbed and stop data collection. In the supplementary material sections A.2, A.3, we provide visual explanations of our validity conditions and figures demonstrating the robustness of the DINO features in detecting varying environment disturbances for an object.

### 3.4 Policy

**Training.** We train a separate policy $\pi$ for each task as an LSTM network with behavioral cloning that receives as input the RGB and force-torque observations in the dataset $\mathcal{D}_{new}$ and regresses the corresponding actions. Note that $\mathcal{D}_{new}$ does not contain proprioception data, allowing our policies to generalize to different object poses naturally due to the use of our wrist camera.

**Inference.** We deploy our policy $\pi$ to solve a task up to the $R_{th}$ demonstrated state. If no environment disturbance occurred during data collection for that task, then the $R_{th}$ state is the final state in the demonstration and $\pi$ solves the task completely in a closed-loop manner. Otherwise, after $\pi$ completes the task up to the $R_{th}$ state, the remaining demonstrated action segment $\zeta_{remaining}$ *is replayed.* More details regarding how we deploy our policy, the network architecture, and how we detect that $\pi$ has reached the $R_{th}$ state can be found in the supplementary material section A.4.

## 4 Experiments

We evaluate MILES through several real-world experiments. Through these experiments, we aim to answer the following questions: 1) Can MILES solve a range of everyday tasks and how does it perform against baselines that learn from a single demonstration? 2) How does MILES perform under different component ablations? 3) How important are vision and force modalities to the performance of MILES? and 4) How does MILES perform under different sizes of self-supervised data? Videos of our experiments can be found on our webpage: www.robot-learning.uk/miles

**Implementation Details.** For our experiments, we use a FLIR camera mounted to the wrist of a Franka Emika Robot. We sample $Z = 10$ augmentation trajectories for each demonstration waypoint ($\approx$ approximately 1 minute of data collection per waypoint). This number is set arbitrarily, but as we show later in our ablations, some tasks may require less data. We collect augmentation trajectories with initial poses near the demonstration in the range of 4cm and 4 degrees around each demonstration waypoint. As commonly done in the literature [3, 16, 17, 20], we provide our demonstrations starting near each object. At deployment, to reach the object from far away we first estimate the object's pose using pose estimation and approach it before switching to MILES. Finally, we set the environment disturbance threshold $\theta$ to 0.94 for all our tasks. Additional details on the pose estimation method we use and how to set each one of MILES' parameters can be found in our supplementary material sections B.1 and B.2.

| Methods | Lock with key | Insert USB | Plug into socket | Insert power cable | Twist screw | Bread in toaster | Open lid | Mean |
|---|---|---|---|---|---|---|---|---|
| Demo Replay | 0 | 0 | 0 | 0 | 0 | 15 | 25 | 6 |
| Reset Free Residual RL | 0 | 15 | 35 | 0 | 0 | 0 | 0 | 7 |
| Reset Free FISH | 0 | 30 | 25 | 15 | 0 | 0 | 0 | 10 |
| Pose Estimation + Demo Replay | 50 | 10 | 85 | 80 | 70 | **100** | **100** | 71 |
| **MILES** | **90** | **70** | 85 | **85** | **85** | 95 | **100** | **87** |

Table 1: Task success rates (%) for 20 trials reported for each method.

## 4.1 Can MILES solve a range of everyday tasks and how does it perform against baselines that learn from a single demonstration?

In this experiment, we assess the performance of MILES across a diverse set of everyday tasks and compare it to various baseline methods capable of learning from a single demonstration. We select seven distinct tasks, shown in Figure 4, spanning a range of complexities, each learned from a single demonstration. The tasks are: 1) Lock with key; 2) Insert USB; 3) Plug into socket; 4) Insert power cable; 5) Twist screw; 6) Bread in toaster; 7) Open lid. Tasks 1-4 are *contact-rich* and and our setup follows prior work on contact-rich manipulation [16, 30, 31, 32, 33] and the NIST benchmark [34]. Similar to prior work [35, 36], we focus our evaluation on single-task performance, but also report results on generalization, robustness to distractors, and multi-stage tasks in section 4.5. We provide a detailed description of each task in the supplementary material section B.3 including data collection times, information on which tasks involve demonstration replay, which tasks stopped data collection due to an environment disturbance and information on the length of each human demonstration.

**Baseline Methods.** We chose 4 baselines that can learn from a single demonstration without prior task knowledge, similar to MILES. (1) **Demo Replay** which involves replaying the demonstrated actions. (2) **Pose Estimation + Demo Replay** follows [17] and leverages MILES' data to perform pose estimation followed by demonstration replay. (3) **Reset Free Residual RL** replays the demonstration's actions at each timestep and learns corrective actions on top using DDPG [37]. Like MILES, no human intervenes to reset the environment during training, hence we call it "Reset Free". Finally, (4) **Reset Free FISH** uses the state-of-the-art RL-based imitation learning method FISH [3] but no human intervenes to reset the environment during training. Further, implementation details on the baselines can be found in our supplementary material section B.4.

**Evaluation.** For a fair evaluation, we carefully tuned each method's hyperparameters. Additionally, each learning-based baseline collected the same number of observations as MILES during data collection for each task. We evaluated each method's success rate across 20 trials. For each trial we randomized the relative starting pose of the robot and the task-relevant object equivalently across all methods within a sphere of 20cm around the object as long as the object was visible to the camera. Finally, we emphasize that for all evaluations both MILES and the baselines predict *6-DoF actions*.

**Results.** Table 1 presents the success rates of MILES and the 4 baselines across 7 tasks. As shown **MILES** obtains a high success rate across all tasks. Specifically, MILES obtains an average of 87% success rate across all tasks with the "Open lid" task having the highest success rate of 100%. From the contact-rich tasks, inserting the key and locking the lock achieved the best performance despite the task's low tolerance and complex interaction, with an impressive 90% success rate. The lowest success rate is observed in the USB insertion task, where MILES obtains 70%, a performance dip we attribute to the task's low tolerance of less than 1mm. Despite the USB task, for the remaining tasks that required high precision MILES was able to complete them consistently well.

The next best-performing method, **Pose Estimation + Demo Replay**, obtained an average success rate of 71%. Our experiments showed that small errors in object pose estimation led to failures due to the compounding errors of demonstration replay, as observed in prior work [2, 20]. This is particularly evident in tasks requiring precise manipulation including "Lock with key", "Insert USB" and "Twist screw". Instead, while MILES also replays part of the demonstration for some tasks, the fact that it does so for a much shorter horizon allows it to obtain considerably higher success rates.

**Reset Free FISH** failed to solve the majority of tasks, yielding an average performance of 10% across all evaluated scenarios. During training, for several tasks, the policy caused significant disturbances to the environment that made policy learning very hard without manual resetting. We believe that this is the central reason behind Reset Free FISH's low performance Additionally, compared to the original FISH implementation, as we train policies that predict 6-DoF actions the learning efficiency of Reset Free FISH is negatively affected. Lastly, **Reset Free Residual RL** obtained a

| Method Ablations | Lock with key | Insert USB | Plug into socket | Insert power cable | Twist screw | Bread in toaster | Open lid | Mean |
|---|---|---|---|---|---|---|---|---|
| No Sequence | 60 | 20 | 20 | 10 | 0 | 85 | 95 | 43 |
| No Environment Disturbance | 90 | 70 | 85 | 85 | 0 | 0 | 0 | 47 |
| No Reachability | 75 | 40 | 95 | 20 | 85 | 95 | **100** | 73 |
| No Memory | 50 | 65 | **100** | 75 | 35 | 90 | **100** | 74 |
| **MILES** | **90** | **70** | 85 | **85** | **85** | 95 | **100** | **87** |

Table 2: Task success rates (%) for 20 trials reported for each method ablation.

lower success rate, averaging 7% due to similar challenges with Reset Free FISH. As anticipated, the **Demo Replay** baseline was the least effective among the baselines. Simply replaying the demonstration without pose estimation or online action correction leads to task failures.

While the RL methods would have benefited from manual environment resets during training, our results demonstrate that in our "easy" imitation learning setting, where only a single demonstration is available without additional human interventions, MILES significantly outperformed the baselines.

**Simulation Evaluation.** Finally, to support the reproduction of our results, we conducted additional experiments on our method and the baselines in simulation using 5 tasks from the RLBench benchmark [38] which can be found in the supplementary material section C.1.

## 4.2  How does MILES perform under different method ablations?

This section studies MILES' performance by ablating 4 different components of the method: (1) **No Environment Disturbance:** we ablate the environment disturbance condition by not checking for that condition when collecting augmentation trajectories. (2) **No reachability:** we ablate the reachability condition by relabeling each observation's action (of the existing MILES data), to move the robot to the nearest waypoint in the demonstration based on their Euclidean distance. If the constraint for reachability is not important, then simply moving from each pose to the nearest waypoint in the demonstration in a straight line would be sufficient to solve a task. (3) **No sequence:** we recollect MILES' data but instead of collecting $Z$ augmentation trajectories for the first demonstration state, then progressing to the second state and so on, we collect data *without* following the demonstration's waypoint sequence and instead follow a random one. (4) **No Memory:** For this ablation we retrain a network on the existing MILES data that does not account for history.

**Results.** Table 2 shows MILES performance after ablating each component. Collecting augmentation trajectories for each demonstration state in a random order (**No Sequence**), with an average success rate of 43%. Additionally, not checking for the environment disturbance condition (**No Environment Disturbance**) appears to cause significant performance degradation for the tasks where an environment disturbance occurred during data collection, corresponding mostly to the non-contact rich tasks. On the other hand, not checking for the reachability condition (**No Reachability**) also lowers performance, particularly for the precise, contact-rich tasks, indicating that the reachability condition is the most important when learning tasks requiring precise manipulation. Finally, the lower performance obtained by removing the LSTM (**No Memory**) demonstrates the performance benefits of training memory-based networks on datasets collected using MILES.

## 4.3  How important are vision and force modalities to the performance of MILES?

In this section, we ablate the use of vision and force feedback as policy inputs for the four contact-rich tasks from our earlier experiments. We retrain and evaluate two policies: one using only vision and one using only force. The results, shown in Figure 5, indicate that the vision-based policy improves MILES' performance in the "Insert USB" and "Plug into socket" tasks but reduces performance in the other two tasks. This suggests that force feedback might not consistently benefit MILES, possibly due to its noisy signal which makes it hard

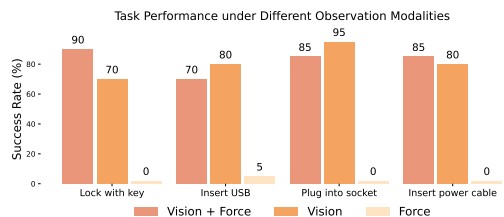

Figure 5: MILES' performance when trained only on either vision or force feedback or both.

to distinguish between different environment states. The force-based policy, however, fails almost completely. This is expected as force feedback is zero in free space and can be ambiguous due to

symmetries in object surfaces. Overall, while force feedback aids performance in some tasks, it is not always necessary. Vision remains the most crucial modality to MILES' high performance.

### 4.4 How does MILES perform under different sizes of self-supervised data?

In this section, we ablate the dataset size used to learn four tasks by splitting their original datasets into chunks containing 75%, 50%, and 25% of the original data. We evaluated the best and worst performing contact-rich tasks ("Lock with key" and "Insert USB") and non-contact-rich tasks ("Open lid" and "Twist screw"). Data collection times for each task can be found in the supplementary material. Figure 6 shows that for high tolerance tasks like "Open lid," MILES achieves a 100% success rate even with 25% of the data,

corresponding to *only* 8 minutes of data collection. However, for precise tasks, success rates decrease as dataset size is reduced. Notably, for "Lock with key" and "Twist screw," reducing the dataset to 50% results in a high failure rate. To summarize, we observe that high tolerance tasks are likely to require less data, and in practice only a few minutes of data collection time. Instead, for high-precision tasks, like inserting a USB, the dataset size appears to impact MILES' performance significantly.

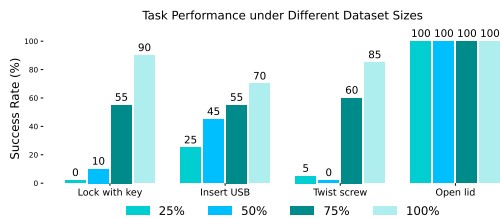

Figure 6: MILES' performance when trained on different dataset sizes. 100% corresponds to the original dataset. 75%, 50%, and 25% correspond to splits of the original dataset.

### 4.5 Further questions about MILES

**Can MILES generalize?** We conduct additional experiments on generalization for MILES in the supplementary material section D.1. **Can MILES perform multi-stage tasks?** We provide additional experimental results on how MILES performs on multi-stage tasks in the supplementary material section D.2. **Is MILES robust to distractor objects?** We conduct additional experiments studying MILES' performance in the presence of distractors in the supplementary material section D.3. **What if MILES stops data collection early due to a detected environment disturbance?** We provide a discussion and intuition on MILES' behavior in scenarios where data collection stops early in our supplementary material section D.4.

## 5 Discussion

**Limitations.** We now highlight some important limitations of our method. Firstly, MILES' reliance on a wrist camera enables MILES to obtain spatial generalization, however, simultaneously this limits its field of view and its applicability to larger task spaces. Future work could address this by incorporating an external camera to initially approach an object before switching to the wrist camera, similarly to [20]. Secondly, while MILES is robust to distractors at deployment before data collection begins it requires a human to set up the robot's workspace such that only the task-relevant object is in camera view for the policy to achieve spatial generalization. While this requires only a few seconds of human time, future work could address this by extending MILES to incorporate segmentation methods, similar to [20, 39], that segment the task-relevant object in the dataset. Similarly, to address any unwanted collisions that MILES could cause in the presence of multiple objects, future work could study incorporating an external camera during self-supervised data collection to plan and collect collision-free augmentation trajectories. Thirdly, our current implementation of MILES trains a separate policy for each task and hence it is unclear how well MILES would generalize to completely new tasks. In future work, we aim to study this by training a single monolithic policy on MILES' self-supervised data combined with replay-trajectory retrieval [40] similarly to our generalization task in section 4.5.

**Conclusion.** We introduced MILES, a framework that makes imitation learning easy. MILES requires only a single demonstration and collects self-supervised data that demonstrate to the robot how to return to and then follow that demonstration. Subsequently, this enabled us to obtain manipulation skills comprising either (1) a single end-to-end policy trained with behavioral cloning, or (2) a combination of an end-to-end policy and demonstration replay. Our real-world experiments showed that when only a single demonstration is available, self-supervised data enable the acquisition of skills that achieve considerably improved performance compared to several state-of-the-art baselines. MILES can learn everyday tasks, ranging from opening a lid, to using a key to lock a lock, to inserting a USB stick into a port, requiring complex and precise contact-rich manipulation.

**Acknowledgements**

We would like to thank Norman Di Palo, Vitalis Vosylius, Pietro Vitiello, Kamil Dreczkowski, Yifei Ren and Ivan Kapelyukh for their insightful discussions on the method and feedback in drafting the first versions of this paper.

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

# Supplementary Material
# MILES: Making Imitation Learning Easy with Self-Supervision

For videos demonstrating MILES' performance and code implementation please see our webpage: www.robot-learning.uk/miles.

## A MILES: Additional Details on the Method

### A.1 Method Pseudocode

We provide a detailed **pseudocode** describing MILES, in Algorithms 1- 8.

### A.2 Validity Conditions for Augmentation Trajectories

As described in section 3.3, after collecting data for an augmentation trajectory, we check for two conditions: (i) **reachability** and (ii) **environment disturbance**, to determine whether an augmentation trajectory is valid and eligible to fuse with the demonstration. Figure A.2 shows examples of these two conditions.

#### A.2.1 How do we check for the Reachability condition?

**Reachability.** To check for reachability, after executing an augmentation trajectory $\tau_k$, we verify whether the final achieved pose matches the pose of the $k_{th}$ demonstration waypoint using proprioception, as described in section 3.3. Pseudocode describing how we check for reachability is also provided in Algorithm 3. It is crucial to check for reachability because an augmentation trajectory that does not meet this condition cannot be fused with the demonstration, as it cannot return to the demonstration state. If the waypoint $w_k^\zeta$ is unreachable during data collection, we cannot automatically determine how to reach $w_k^\zeta$ from $w_M^{\tau_k}$, without collecting observations that do so during self-supervised data collection. Consequently, we cannot automatically determine what actions to take to return back to the demonstration from $w_M^{\tau_k}$, as we can with valid augmentation trajectories. Figure A.2 (a, left) shows an example where the reachability condition is not met due to environmental dynamics, such as a key getting "jammed" and failing to reach the target waypoint due to collision and friction in the lock. A similar example where the reachability condition is met is shown in Figure A.2 (a, right).

#### A.2.2 How do we check for the Environment Disturbance condition?

**Environment Disturbance.** To determine whether an environment disturbance occurred, we compare the RGB image captured at the $k_{th}$ demonstration timestep with the RGB image captured at the final timestep of the augmentation trajectory, as described in section 3.3. A detailed pseudocode describing how we determine whether an environment disturbance occurred can be found in Algorithm 5, and a visual example can be seen in Figure A.2 (b). The comparison between the two RGB images relies on the similarity of their DINO features [28]. Specifically, we use a pre-trained DINO ViT [28] to obtain the DINO features for different patches of each image similarly to [29]. By computing the cosine similarity between the DINO features of each corresponding image patch in $I_k^\zeta$ and $I_M^{\tau_k}$, we can calculate the average similarity between the two images [29]. If the similarity is below a threshold $\theta$ (to see how we automatically determine $\theta$ please see section B.2.3), we assume the robot has disturbed the environment, and data collection is stopped. Our experiments showed that DINO ViT features are necessary because they are robust to lighting changes and noise in the RGB image. Other methods we tried, such as template matching or computing the per-pixel Euclidean distance, proved brittle and sensitive to lighting variations or noise in the captured images. Understanding why checking for an environment disturbance is important is straightforward. Consider the rectangular object shown in Figure A.2 (b), and assume the task is to learn how to pick up that object. If the robot pushes the rectangular object, causing it to fall over during data collection, the image observed after returning to the demonstration state will no longer match that state's observation from when the demonstration was provided. Consequently, from the point where the disturbance occurred onward, we have no way of knowing how to reach any of the remaining demonstration

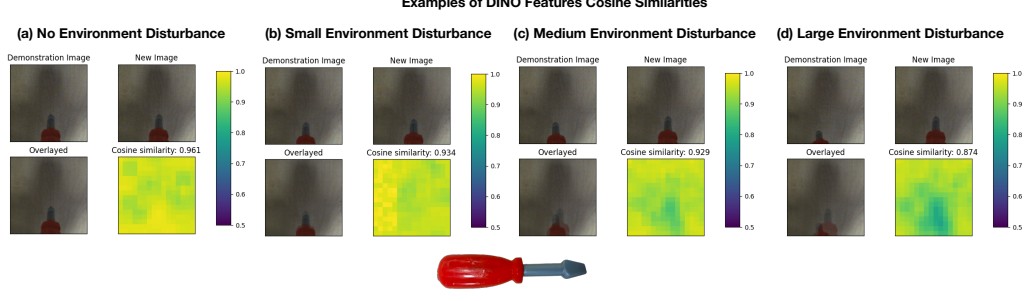

Figure A.1: The cosine similarity computed using the DINO features for the screwdriver task under varying environment disturbances.

states and as a result how to solve the task. This is because we only know how to solve a task by learning how to follow the demonstration after returning to it. But if an environment disturbance has occurred (e.g., the rectangular object has fallen), following the demonstration's actions no longer leads to task completion. Hence, if data collection continued, all future augmentation trajectories would contain invalid observations and actions, as they would demonstrate behavior that does not solve the task that the human demonstrated. This is why we stop data collection after detecting an environment disturbance.

## A.3   Additional Results on Environment Disturbances and DINO Features

We demonstrate in this section several examples of possible environment disturbances and how we can detect them using the DINO features on the toy screwdriver used in our experiments. We use the screwdriver as an example as during data collection for the "Twist screw" task, data collection was stopped due to an environment disturbance caused at the grasped screwdriver. Additionally, disturbances caused in the grapsed objects are often the most subtle, and as such make for the most interesting cases.

Figure A.1 (e) shows the screwdriver object (not grasped). All the other figures depict the screwdriver as it appears in the view of the wrist camera when grasped by the robot. Figure A.1 (a) shows a "Demonstration Image" and a "New Image" that depicts the DINO Cosine similarity (higher better) when no environment disturbance has occurred, i.e., the grasp has not changed. The heatmap demonstrates the similarity between each corresponding patch between the "Demonstration Image" and the "New Image" (the cosine similarity reported is the mean of these). As shown, the cosine similarity (0.961) is greater than our universal threshold $\theta$ of 0.94 (for more details please see experiments section 4). The reason it is not a perfect 1.0 is due to noise and light changes as the photos were captured at different moments in time. Figure A.1 (b), shows a detected environment disturbance based on the DINO features. As shown under the "New Image" the screwdriver has moved by a small amount in the gripper and the cosine similarity falls slightly below our threshold $\theta$. Then, Figure A.1 (c) shows a slightly bigger detected environment disturbance, and finally Figure A.1 (d) shows a rather large environment disturbance. Generally, as shown in Figure A.1, the DINO features are robust in detecting environment disturbances of different scales and as we move from smaller to larger disturbances in the grasped screwdriver the cosine similarity also decreases, as expected.

## A.4   MILES' Policy

**Training:**   To train our manipulation policy we leverage the dataset $\mathcal{D}_{new}$ comprising the fused augmentation trajectories with the demonstration as described in section 3.4. MILES' policy $\pi$ comprises either (1) an end-to-end network trained with behavioral cloning (BC) or (2) an end-to-end network trained with BC combined with demo replay, which is utilized when data collection was interrupted due to a detected environment disturbance.

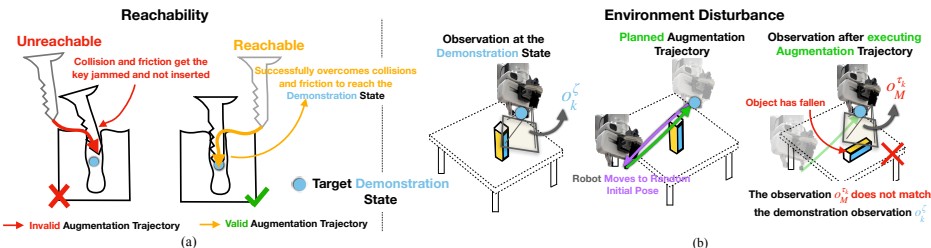

Figure A.2: **Reachability:** Two examples of possible augmentation trajectories for a locking task are shown; an invalid trajectory (left) that fails to reach the target demonstration waypoint due to collisions, friction, and potentially inevitable systematic controller errors and a valid one (right) that successfully reaches the target waypoint. **Environment Disturbance:** As the robot collects an augmentation trajectory, it perturbs the environment such that after returning to the demonstration's waypoint the live observation and the demonstrated one no longer match, indicating that data collection should stop.

### A.4.1 How is our policy defined when No Environment Disturbance occurred during data collection?

**No Environment Disturbance.** When no disturbance occurred our dataset $\mathcal{D}_{new}$ contains augmentation trajectories that can return to and then follow the demonstration from every state. In that case, we leverage $\mathcal{D}_{new}$ to train an end-to-end behavioral cloning policy $\pi$ that comprises a single neural network $f_\psi$, parameterized by $\psi$, that receives as input an RGB image captured from the wrist camera and force-torque feedback to predict 6-DoF actions: $f_\psi : \mathbb{R}^{H \times W \times 3} \times \mathbb{R}^6 \rightarrow SE(3)$ as well as an additional binary value indicating the gripper action ($\mathbb{R}^{H \times W \times 3}$ refers to the RGB images where $H$: height, $W$: width and $\mathbb{R}^6$ to measured forces and torques). The force-torque feedback is captured directly using Franka Emika Panda's joint force sensors. For our policy to generalize spatially, *no* proprioception input is passed to $f_\psi$ and all actions are predicted relative to the EE's frame. $f_\psi$ consists of a ResNet-18 backbone [41] for processing RGB images, and a small MLP embeds force feedback into a 100-dimensional space. The output of the force MLP and ResNet-18 are concatenated and fed into an LSTM [42] network for action prediction. The network is trained using standard behavior cloning to maximize the likelihood of $\mathcal{D}_{new}$.

### A.4.2 How is our policy defined when an Environment Disturbance occurred during data collection?

**Environment Disturbance.** When self-supervised data collection was stopped due to an environment disturbance, our dataset $\mathcal{D}_{new}$ contains augmentation trajectories that can return the robot to any state from the initial demonstration state up to the demonstration state at timestep $R$, where $R < N$ (see section 3.2). In this scenario, if our policy consists only of $f_\psi$, then during task execution the robot would be able to solve the task only up to the $R_{th}$ state, but not complete it. As such, we define our policy $\pi$ to consist of two components: (1) the first component is a neural network $f_\psi$ identical to the above scenario, but trained up to the $R_{th}$ state and (2) the second component corresponds simply to the sequence of the remaining demonstration actions from the $R_{th}$ state onwards, for which no self-supervised data was collected, i.e., $\zeta_{remaining} = \{a_n^\zeta\}_{n=R}^N$.

### A.4.3 How do we deploy MILES' policy?

**Deployment:** Our LSTM-based policy closely follows the implementation of BC-RNN [43]. Deploying the policy is straightforward and depends on whether data collection was interrupted due to an environment disturbance. If uninterrupted, then only the neural network $f_\psi$ is used to complete the task equivalently to policies trained using reinforcement learning or behavioral cloning.

If data collection was interrupted, first $f_\psi$ is deployed to solve the task up to the $R_{th}$ state in an identical way as the scenario of "no environment disturbance". After the robot reaches the $R_{th}$ state then $\zeta_{remaining}$ is executed. We determine whether the closed-loop policy has completed the task up to the $R_{th}$ in a very simple way as described in section A.4.4.

During deployment we reset the hidden state of the LSTM at an interval equal to two times the number of timesteps (i.e., waypoints) in the demonstration for which augmentation trajectories were collected. For example, if for a task MILES collected augmentation trajectories for 40 demonstration

waypoints before stopping due to an environment disturbance, then, during deployment the hidden state of the LSTM is reset every 80 timesteps. We did not find the frequency of resetting the hidden memory to have significant effects on the policy's performance. We would like to note that the only important observation we made was that the number of timesteps should not be very low (e.g., 5) as then the robot would end up progressing towards completing a task very slowly.

Pseudocode describing MILES' policy deployment can be found in Algorithm 8.

### A.4.4 How do we determine when to switch from closed-loop control to demonstration replay?

Switching from closed-loop to demonstration replay is straightforward. As the objects and the robot can be at different poses during deployment from the ones during data collection, we cannot just use the robot's proprioception to know when the $R_{\text{th}}$ state has been reached. Hence, we deploy $f_\psi$ until it predicts continuously the identity transformation, indicating no robot movement. Then, we switch to demonstration replay, where we replay the rest of the demonstration $\zeta_{remaining}$.

## B  More details on the Experimental Setup

### B.1  Pose Estimation

In practice, as with most methods [3, 16, 17, 20], we naturally provide the demonstrations starting near the task-relevant object to focus self-supervised data collection at the part of the task that is the most important, that is the robot-object interaction part.As such, we need a way to ensure that MILES can still solve any task regardless of how far the robot is from an object. An apparent solution to this is to provide the demonstration starting from a pose far away from the object and deploy MILES' data collection. While this is possible – as MILES makes no assumptions or restrictions on the length of the demonstration– it may be inconvenient. As such, inspired by [2, 18, 20] we use a simple pose estimator at deployment to estimate the relative pose between the robot at the initial state of the demonstration (for which MILES collected data) and the task-relevant object. As we do not assume any 3D object models, we use the method deployed in [17] although any other model-free pose estimator can be used. This allows us to first coarsely estimate the pose and move near the task-relevant object from any robot starting pose before deploying MILES. Uncut videos demonstrating this behavior can be found on our webpage: www.robot-learning.uk/miles.

### B.2  MILES Data Collection Hyperparameters

#### B.2.1  How do we set the data collection range around each demonstration waypoint?

As discussed in our experiments section 4, we collect data in a range of 4cm and 4 degrees around each demonstration waypoint. However, this range is *not limiting* and can be set to *any desirable range* like any other robot learning method. In our case, we set this range to be the average pose estimation error to reach the initial pose of the demonstration relative to the task-relevant object using the pose estimation method described in section B.1 which we obtained based on [17].

#### B.2.2  How do we determine the number of augmentation trajectories to collect for each demonstration waypoint?

For all of our experiments, we set the number of augmentation trajectories per demonstration way-point, $Z = 10$. In our case, we set this arbitrarily, but as we showed in our method's data collection ablation in section 4.4 different tasks require different numbers of augmentation trajectories. As such, we provide two guidelines for setting the value for $Z$. Firstly, high tolerance tasks, like the "Open lid" task reported in our experiments usually require a small number of augmentation trajectories. On the other hand, precise tasks, like the "USB insertion" task reported in our experiments require more augmentation trajectories. Secondly, as the data collection range around each demonstration waypoint increases, the number of augmentation trajectories collected should also increase with an approximately linear relationship, i.e., if the range is doubled, then the number of augmentation trajectories should be doubled as well. We recommend as a starting point, for a data collection range similar to our experimental setting of 4cm and 4 degrees, to collect 10 augmentation trajectories for precise, low-tolerance tasks, and 4 augmentation trajectories for high-tolerance tasks.

| Task: | Description | DCT | Task: | Description | DCT |
|---|---|---|---|---|---|
| Lock with key | Insert a key into a lock and rotate 90 degrees to lock it. | 24' | Twist screw | Insert a toy screwdriver into a screw and twist by 90°. | 22' |
| Insert USB | Insert a USB stick into a USB port (< 1mm tolerance) | 21'. | Bread in toaster | Put a plastic bread inside a toaster. | 40' |
| Plug into socket | Plug a UK plug (3-pin) to a socket. | 37' | Open lid | Lift the lid of a blue box. | 31' |
| Insert power cable | Plug the power cable into the power port of a PC. | 28' | | | |

Table 3: Task descriptions of the 7 tasks used in our experiments. **DCT** stands for Data Collection Time and corresponds to the time spent collecting self-supervised data.

### B.2.3  How do we determine the Environment Disturbance threshold $\theta$ ?

We determined $\theta$ simply by spawning several random RLBench [38] tasks in CoppeliaSim and running MILES. By setting up custom heuristics that determine environment resets in the simulation we found that for the DINO model we use, a similarity of $\theta < 0.94$ appeared to detect environment disturbances across all tasks successfully. Consequently, we used that in our real-world experiments too.

### B.3  Task Descriptions

A detailed description of each task along with their Data Collection Times (**DCT**) can be found in Table 3.

### B.3.1  How long is each demonstration?

The demonstration lengths varied across each task. As follows, we list for each task the number of demonstration waypoints comprising each human demonstration (each demonstration waypoint can be interpreted as a timestep): Lock with key: 32, USB task: 20, Plug into socket: 40, Insert power cable: 29, Twist screw: 47, Bread in Toaster: 70, Open lid: 80. All demonstrations were collected using teleoperation. Note that the number of demonstration waypoints is not necessarily equal to the number of waypoints for which MILES collected augmentation trajectories. This is because environment disturbances may have caused the data collection to stop earlier.

### B.3.2  For which tasks was an Environment Disturbance detected?

An environment disturbance was detected for the following tasks: `Twist screw`, `Bread in Toaster` and `Open lid`. As such for these tasks the policies comprise a closed-loop and a demonstration replay component.

We also note that for the lock with key task, we stopped data-collection "half-way" through the 90 degrees twisting rotation for hardware safety. This is because the forces exerted on the robot as it was collecting self-supervised data were too high. In this case, we treated this identically to an environment disturbance. At deployment, the learned policy completes most of the task closed-loop, apart from a small twisting motion done with demo replay, after the closed-loop policy converges to predicting the identity transformation as discussed in section A.4.4. This is similar to adding force limits to reinforcement learning algorithms and was done to protect our robotic hardware; however, doing so is not a requirement.

### B.4  Baselines

Here, we provide further implementation details on two of the baselines we used in our paper.

**Pose Estimation + Demo Replay.** For this baseline, we follow the same problem formulation as in [17], but improve upon that baseline in two key ways: (1) the data on which it is trained on is the same data collected for MILES, as such it contains only valid trajectories that cover a larger part of the task space and (2) instead, of replaying recorded velocities, we also replayed the recorded forces which is particularly important for the contact rich tasks. This baseline estimates and moves the

| Methods | Insert Onto Square Peg | Lightbulb In | Pick Up Cup | Turn Tap | Lamp On | Mean |
|---|---|---|---|---|---|---|
| Demo Replay | 0 | 0 | 5 | 5 | 0 | 2 |
| Reset Free Residual RL | 0 | 0 | 0 | 0 | 0 | 0 |
| Reset Free FISH | 0 | 0 | 0 | 0 | 20 | 4 |
| Pose Estimation + Demo Replay | 70 | 65 | 90 | **80** | 95 | 80 |
| **MILES** | **90** | **75** | **100** | 75 | **100** | **88** |

Table 4: Task success rates (%) of each method on RLBench.

robot to a pose relative to the object of interest as depicted in the first state in the demonstration and replays the complete demonstration. We chose this baseline compared to alternatives, as it leverages task-specific data allowing it to achieve very precise pose estimation.

**Reset-Free FISH** [3]. For Reset-Free FISH we use the implementation provided by the authors as it can be found in: https://github.com/siddhanthaldar/FISH. We only changed the implementation such that the policy always predicts 6-DOF actions instead of constraining the output to specific DOFs, as doing so assumes access to prior task knowledge. To learn residual actions on top of the demonstration we tested both using demo replay as the base policy, as well as VINN [44] but found that demo replay led to better performance.

## C  Additional Experiments

### C.1  Simulation Results

To aid other researchers in reproducing our results, we conducted additional simulation experiments on the RLBench benchmark [38] on 5 tasks, specifically: 1) 'Insert Onto Square Peg', 2) 'Lightbulb In', 3) 'Pick Up Cup', 4) 'Turn Tap' and 5) 'Lamp On'. We performed an identical evaluation to our real-world experiments where we performed 20 evaluation trials for each method. Additionally, we used the images captured only from the wrist camera in RLBench. During training we allowed each method to collect the same amount of data and **we did not perform any environment resets during training/data collection for any methods**. The results can be seen in Table 4. As shown, MILES significantly outperforms the baselines, while the relative performance when comparing all methods remained relatively unchanged compared to our real-world results.

Similarly to our real-world experiments, the reinforcement learning baselines obtained poor performance for reasons in line with the ones discussed in our experiments section 4.1. Specifically, during training we observed that for the tasks 'Insert Onto Square Peg' and 'Lightbulb In' a random gripper action drops the grasped object during exploration and the policy never manages to grasp it again during training without a reset in the given training time. For the 'Pick Up Cup' task, the reinforcement learning policy knocks the cup off the table during exploration, consequently never learning something useful. For the 'Turn Tap' task the RL policies never learned to properly grasp and rotate the handle and for the 'Lamp On' task, only Reset Free FISH managed to learn a policy that obtains 20% success rate in the given training time. As discussed in our real-world experiments, if instead we had allowed environment resets and more training time that would have resulted in significantly higher success rates for the RL baselines, compared to their current performance.

## D  Further questions about MILES

### D.1  Generalization Performance

Since MILES uses BC to train policies, existing generalization results for BC [9, 1] also apply to MILES. For tasks that include demonstration replay following the closed-loop policy, MILES can generalize to new objects by retrieving the replay trajectory of the most similar object in the existing demonstrations, similar to prior work [19]. To test this, we tasked MILES with throwing markers of different colors into differently shaped and colored bins, shown in Figure 4 (8). Trained on five bins (marked green) and tested on two new bins (marked red), MILES achieved an 80% success rate on the pink bin and 60% on the gray bin, over 10 trials each starting from poses where simple demonstration replay would fail. The data collection time for this task was on average 34 minutes for each bin and an environment disturbance was detected for each bin. To determine which remaining actions to replay for the previously unseen bins, we selected the remaining actions from the bin in

the training set whose RGB image in the demonstration has the highest similarity in terms of DINO features with the bin during deployment, inspired by prior work [19]. Videos exhibiting MILES generalization on the two test case bins can be found on our webpage: www.robot-learning.uk/miles.

## D.2 Multi-stage Tasks

To evaluate MILES' ability to solve multi-stage tasks, we tasked MILES with picking up the plastic bread shown in Figure 4 (6) (as part of the "Bread in Toaster" task) and inserting it into the toaster. To achieve this we broke the task down into two stages: first, we provided a demonstration showing how to pick up the bread and trained MILES. Then, we used the policy already trained on the "Bread in Toaster" task to finish the task. To link the two stages together, first the policy to pick up the bread is deployed. After, the execution ends, the robot returns to its default position. Then, the pose estimation method described in section B.1 is deployed to approach the toaster, and then the policy trained with MILES is deployed to insert the bread into the toaster. Videos exhibiting MILES' multi-stage task performance on picking up and inserting the bread into the toaster can be found on our webpage: www.robot-learning.uk/miles.

## D.3 Performance with distractors

We found that performing standard image augmentation techniques, including changing the brightness, contrast, noise, cropping random image parts, etc. allowed MILES to be robust to distractor objects, as shown in the videos provided on our webpage: robot-learning.uk/miles.

## D.4 What if MILES stops data collection early due to a detected environment disturbance?

There is no requirement as to how early MILES may stop data collection due to an environment disturbance, as long as it has collected sufficient augmentation trajectories for at least the first demonstration waypoint. During data collection, MILES can effectively learn a policy even if an environment disturbance occurs early. Unlike RL, MILES learns to solve the task closed-loop up to the demonstration waypoint where the disturbance was detected, after which it replays the demonstration. This is because MILES collects data progressively for each demonstration waypoint, rather than rolling out a policy all at once like RL. Consequently, during data collection, if a disturbance occurs as early as (for example) near the 2nd waypoint, MILES will still know how to get to the 1st waypoint during deployment, where it will replay the demonstration.

Overall, MILES can handle early environment resets during data collection. While as with the majority of learning-based methods, the more the data the better the performance, as such the later an environment disturbance occurs in the data collection process the better. However, MILES can still learn a robust policy as long as sufficient data has been collected at least for the 1st demonstration waypoint. This is typically trivial as most human demonstrations naturally begin by controlling the robot in free-space far from the object of interest, before interacting with it.

# E    Detailed Pseudocode

---

**Algorithm 1: MILES Overview** (Simplified)

---

**Input:** Single Task Demonstration: $\zeta = \{(w_n^\zeta, o_n^\zeta, a_n^\zeta)\}_{n=1}^N$, Number of augmentation trajectories per demonstration waypoint $Z$, environment disturbance threshold $\theta$ (Default: $\theta = 0.94$)

1: $\mathcal{D} = \{\}$ // `init empty dataset of augmentation trajectories`
2: Reachable = **True** // `init variable that tracks reachability`
3: Disturbance = **True** // `init variable that tracks environment disturbances`
4: $R = 1$ // `init variable that stores the timestep when self-supervised data collection stops`
5: Move robot to the initial demonstration pose $w_1^\zeta$
6: **for** iteration $k = 1$ to $N$ **do**
7:     $j = 1$ // `init variable that tracks the number of collected augmentation trajectories per demo waypoint`
8:     **while** $j \leq Z$ **do**
9:         $\tau_k \leftarrow SampleTrajectory(w_k^\zeta)$ (Alg. 2)
10:        Reachable $\leftarrow CheckReachability(w_k^\zeta)$ (Alg. 3)
11:        **if** Reachable is **False then**
12:            $ReturnToDemoWaypoint(k, \zeta)$ (Alg. 4)
13:            Break // `exit while loop`
14:        **end if**
15:        $I_M^{\tau_k} \leftarrow$ Capture RGB wrist-cam image // $M$ `is the` $M_{th}$ `(final) timestep of` $\tau_k$
16:        Disturbance $\leftarrow CheckEnvDisturbance(o_k^\zeta, I_M^{\tau_k}, \theta)$ (Alg. 5)
17:        **if** Disturbance is **True then**
18:            $R = k$ // `store timestep when data collection stops`
19:            Break // `exit while loop`
20:        **end if**
21:        $\mathcal{D} = \mathcal{D} \cup \tau_k$ // `add augmentation trajectory to dataset`
22:        $j = j + 1$
23:    **end while**
24:    **if** Disturbance is **True then**
25:        Break // `exit for loop`
26:    **end if**
27:    Proceed to the next demonstration state by performing action $a_k^\zeta$ // `follow the demonstration's progression`
28: **end for**
29: $\mathcal{D}_{\text{new}} \leftarrow FuseAugmentationsWithDemo(\mathcal{D}, R, \zeta)$(Alg. 6)
30: $\pi \leftarrow TrainPolicy(\mathcal{D}_{\text{new}}, R, \zeta)$(Alg. 7)
31: $Deploy(\pi, R, \zeta)$(Alg. 8)

**Output:** $\pi$

---

**Algorithm 2: SampleTrajectory**

**Input:** Demonstration waypoint $w_k^\zeta$
1: $\tau_k = \{\}$ // init empty augmentation trajectory
2: Sample initial pose $w_1^{\tau_k}$ randomly and move there in a straight line. (Optional: record trajectory poses) // Note that the straight line trajectory may altered due to the collisions and the robot's compliance with its environment.
3: Move back to $w_k^\zeta$ // either by tracking the recorded trajectory poses backward or by re-planning a new, straight-line trajectory (equal performance).
4: $m = 1$ // observations, actions index
5: **while** moving to $w_k^\zeta$ **do**
6:    $\tau_k = \tau_k \cup (w_m^{\tau_k}, o_m^{\tau_k}, a_m^{\tau_k})$ // add waypoints, observations and actions to augmentation trajectory; actions are automatically inferred as the relative EE poses between consecutive timesteps.
7:    ($o_m^{\tau_k}$ comprises wrist cam RGB images + force-torque readings)
8: **end while**

**Output:** Return augmentation trajectory $\tau_k$

---

**Algorithm 3: CheckReachability**

**Input:** Demonstration waypoint $w_k^\zeta$
1: Reachable ← **True** // init reachability variable
2: $w_M^{\tau_k}$ ← EE pose // achieved after executing the augmentation trajectory (comprising $M$ timesteps); read from proprioception
3: Reachable = ($w_M^{\tau_k} == w_k^\zeta$) // check whether poses are equal (within the controller's feasible precision)

**Output:** Reachable

---

**Algorithm 4: ReturnToDemoWaypoint**

**Input:** Demonstration timestep $k$, single demonstration $\zeta$
1: Move to initial demonstration waypoint $w_1^\zeta \in \zeta$ // replay demonstration up to the $k_{th}$ timestep
2: **for** iteration $t = 1$ to $t = k$ **do**
3:    Perform action $a_t^\zeta \in \zeta$
4: **end for**

---

**Algorithm 5: CheckEnvDisturbance**

**Input:** Demonstration observation $o_k^\zeta$, captured live image $I_M^{\tau_k}$, similarity threshold $\theta$
1: Disturbance ← **False** // init environment disturbance variable
2: $I_k^\zeta \in o_k^\zeta$ // retrieve RGB image $I_k^\zeta$ from the demonstration's observations
3: $[f_{I_k^\zeta}^1, f_{I_k^\zeta}^2, ...]$ ←DINO-ViT($I_k^\zeta$) // compute DINO-ViT features [29, 28] for **each** image patch $f_{I_k^\zeta}^x$ for the demo waypoint image
4: $[f_{I_M^{\tau_k}}^1, f_{I_M^{\tau_k}}^2, ...]$ ←DINO-ViT($I_M^{\tau_k}$) // compute DINO-ViT features [29, 28] for **each** image patch $f_{I_M^{\tau_k}}^x$ from the current live environment image (captured after executing the augmentation trajectory).
5: $sim =$AvgCosineSimilarity($[f_{I_k^\zeta}^1, f_{I_k^\zeta}^2, ...], [f_{I_M^{\tau_k}}^1, f_{I_M^{\tau_k}}^2, ...]$)
6: **if** $sim < \theta$ **then**
7:    Disturbance ← **True**
8: **end if**

**Output:** Disturbance

---

**Algorithm 6: FuseAugmentationsWithDemo**

**Input:** Dataset of augmentation trajectories $\mathcal{D}$, final data collection time step $R$, single demonstration $\zeta$
1: $\mathcal{D}_{new} = \{\}$ // init empty dataset to store fused trajectories
2: **for** $\tau_k$ in $\mathcal{D}$ **do**
3:    $\zeta_{segment} = \underbrace{\{(w_n^\zeta, o_n^\zeta, a_n^\zeta)\}_{n=k}^R}_{\text{demonstration segment from } k_{th} \text{ demo waypoint to } R_{th}} \in \zeta$
4:    $\tau_{k_{new}} := \tau_k \cup \zeta_{segment}$
5:    $\mathcal{D}_{new} = \mathcal{D}_{new} \cup \tau_{k_{new}}$
6: **end for**

**Output:** $\mathcal{D}_{new}$

---

**Algorithm 7: TrainPolicy**

**Input:** Dataset of augmentation trajectories + demo $\mathcal{D}_{new}$, final data collection timestep $R$, single demonstration $\zeta$
1: Train neural network $f_\psi$ on $\mathcal{D}_{new}$ using standard behavioral cloning// **Discard** proprioception waypoints ($w_m^{\tau_k}$ and $w_n^\zeta$), only observation inputs are used for $f_\psi$
2: **if** $R < $ length($\zeta$) **then**
3:    $\pi = \{f_\psi, \{a_n^\zeta\}_{n=R}^N\}$ // policy consists of an end-to-end neural net + demo replay (if an environment disturbance stopped data collection before the last demo waypoint)
4: **else**
5:    $\pi = \{f_\psi\}$ // policy consists only of an end-to-end neural net
6: **end if**

**Output:** $\pi$

---

**Algorithm 8: Deploy**

**Input:** Policy $\pi$, final data collection timestep $R$, single demonstration $\zeta$
1: Capture observation $o$ // comprising RGB wrist cam image + force-torque feedback
2: Action $a = f_\psi(o)$
3: Perform action $a$
4: **while** $a$ is not the identity transformation **do**
5:    Capture observation $o$
6:    Action $a = f_\psi(o)$
7:    Perform action $a$
8: **end while**
// if an environment disturbance stopped data collection before the last demo waypoint
9: **if** $R < $ length($\zeta$) **then**
10:    Replay remaining demo $\{a_n^\zeta\}_{n=R}^N$
11: **end if**

