# OpenReview forum: "MILES: Making Imitation Learning Easy with Self-Supervision"
_robot-learning.org/CoRL/2024/Conference — CoRL 2024_

### Official Review · Reviewer_dHST · 2024-07-15
**Impressive results, intuitive method,  some unstated assumptions.**

**Originality:** 4
**Technical Quality:** 3
**Clarity Of Presentation:** 5
**Potential Impact:** 3
**Recommendation:** 4
**Confidence:** 4

**Review:**

## Strengths

**Writing:** The paper is very well written; I was able to thoroughly understand everything with ease.

**Method:** The method is intuitive and combines many techniques to make one algorithm, MILES, which can teach policies given 1 demonstration.

**Experiments:** Real-robot experiments are quite comprehensive and demonstrate that MILES works well, especially against a SOTA 1-demonstration baseline (FISH).

**Supp**: Videos are nice! Appendix answers some important questions.

## Weaknesses

**Missing/Difficult to find information:** A couple things I wish the paper made much clearer in the main paper/experiments section. I searched through the appendix (skimmed through, not extremely thoroughly) but didn’t find this info:

- What is $R$ for each task?
- What are the demonstration lengths for each task?

**Heavy unstated assumptions:** Related to the above points about $R$ and the demonstration lengths, it seems that the tasks are relatively short horizon: single item pick and place or insertion tasks where the robot starts really close to the insertion location with the object already in hand. Supporting this is how the “Demo Replay” baseline can get 25% success rates on Open lid (which shouldn’t be too hard) but also even 15% on Bread in Toaster, which should be quite difficult. With longer horizons, errors would accumulate and the demo replay should be getting 0% across the board. Meanwhile, the intro and abstract sell MILES as a general method that would work everywhere with just 1 demonstration. This could at least be discussed in the limitations section, or the authors can prove me wrong with additional experiments.

- Additionally, the method would fail in cluttered settings where many type of straight line movement/deviations would cause the robot to knock things over and greatly slow down self-supervised data collection.
- Tasks like opening drawers/cabinets would also be difficult as the gripper would be holding onto the item to be opened while the method tries to pull the arm away to create new trajectories; the final image would look very different from the demonstration image and this would also make self-supervised data collection very difficult.

**Minor Issues:**

- Figure 2 is too long and the text is too small, yet, it has lots of wasted white space. Please make a figure that fits within the margins and uses all the available white space to make the text larger? Perhaps it can be simplified a bit too.
- Table 1 and 2 also extend beyond the margin…. simply wrapping in a `\resizebox{}` can fix this.

---------------
AFTER REBUTTAL: I have now updated to a strong accept.

**Quality Of The Limitations Section:**

2

**Questions For Rebuttal:**

See above weaknesses.

**Robotics Focus:**

4

**Summary Of Paper:**

This paper proposes MILES, a single-demonstration imitation learning method which collects its own data with just 1 environment reset to train a policy to solve the intended task. The paper demonstrates results on a variety of precise insertion tasks.

**Summary Of Recommendation:**

Weak accept because I believe the paper should fully clarify the assumptions and make some important information regarding the experimental settings easier to find.

---

### Official Review · Reviewer_onvu · 2024-07-17

**Originality:** 3
**Technical Quality:** 3
**Clarity Of Presentation:** 4
**Potential Impact:** 3
**Recommendation:** 3
**Confidence:** 4

**Review:**

**Strengths**

The problem setting that the paper studies is of great interest and well motivated. Environment resets are costly in the real world, as are collecting demonstrations, so being able to use a single environment reset and single demonstration is compelling. The method is simple and easy to understand, and the paper is well-written and easy to follow. The results showcase the effectiveness of the proposed approach.

**Weaknesses**

The paper could benefit from a more complete and upfront discussion of assumptions and limitations. Some of the items mentioned in the "Setup" section (3.1) seem like they should be made into explicit assumptions, and might also be limitations of the approach. For example, the use of a wrist camera seems necessary for spatial generalization, and the object that must be manipulated must be in the view of the wrist camera at the start of interaction. Furthermore, there are likely additional assumptions -- for example, there shouldn't be other objects in view of the wrist camera (otherwise, this would prevent spatial generalization). One more example is the one in Figure 2 (3.2). When the augmentation trajectory makes contact with objects, the key could move in-hand when making contact with the lock - this would permanently change the state of the environment, and likely require a manual human reset. The only reason it does not is because the key seems rigidly attached to the robot hand.

Other limitations of the approach are also not clear. For example, some kinds of environment disturbances, might be difficult to detect via DINO features - for example a subtle in-hand orientation change, such as the key in the locking task. It also seems like the value of force-torque data in this framework might be limited. Naively, it seems like data collection would stop at the first state where significant contact occurs, in which case replay would be used at inference. Figure 4 also suggests that force-torque data may not be strictly necessary.

One other concern is the range of evaluations presented in the paper. It would be great to see results on common simulation benchmarks to showcase the generality of the approach, to provide a more thorough comparison to other common imitation learning datasets and tasks, and to enhance the reproducibility of the proposed approach. Some suggestions include the robomimic datasets (https://robomimic.github.io/) or RLBench (https://sites.google.com/view/rlbench).

**Post-Rebuttal**: I have updated my score based on the additional discussion with authors and additional results that have been provided.

**Quality Of The Limitations Section:**

1

**Questions For Rebuttal:**

In addition to the points raised in the weaknesses above, I have further questions:

- How does the robot attempt to move back from the random pose to the pose in the dataset during self-supervised data collection? More details on the controller used here, and how the linear path is broken up into robot control commands would be useful.

- Why is MILES better than pose estimation + demo replay - that seems like it should be about as good. Would pose estimation + demo replay obtain higher success by using a bottleneck pose closer to the object of interest? What are some examples of MILES' shorter horizon that provide an advantage over the pose estimation + demo replay baseline?

- It would be great to get more details on the LSTM policy architecture and how it consumes observations at train-time and inference. One common LSTM-based policy is BC-RNN (https://arxiv.org/abs/2108.03298) which unrolls an LSTM policy during a rollout sequentially at each timestep (the hidden state is updated while playing each action) and then reset periodically every N steps. Another approach is to use an explicit observation history with frame stacking, and to unroll the LSTM completely over the observation history window to produce a final action. Which approach is being used here?

**Robotics Focus:**

4

**Summary Of Paper:**

This paper proposes a method for imitation learning that uses a single demonstration and a single environment reset. It collects data in a self-supervised way by sampling poses around the demonstrated trajectory, moving to the sampled pose, and then storing data that shows how to return from the sampled pose to the demonstration trajectory. The method is shown to outperform alternatives on a set of real-world manipulation tasks given the constraints of a single demonstration and single environment reset.

**Summary Of Recommendation:**

The paper presents a simple and effective approach, but more evaluation (especially on settings that can be reproduced by others) and discussion on assumptions and limitations would improve the quality of the paper greatly.

---

### Official Review · Reviewer_3nym · 2024-07-20
**Simple but effective idea, but it may benefit from some clarification**

**Originality:** 3
**Technical Quality:** 4
**Clarity Of Presentation:** 4
**Potential Impact:** 3
**Recommendation:** 3
**Confidence:** 3

**Review:**

Overall, the idea looks sound and straightforward. It is simple at its core, but it works. It is evaluated well through multiple analyses. But I have two clarity questions. During training, you emphasized there are no resets; my understanding is that this means:

[1] The human demonstrates a trajectory, and this is recorded.

[2] The human or the robot resets the scene, but the scene is the same as the demonstrated one (e.g., the manipulated object is precisely in the same pose).

[3] As the joint/pose trajectory that the robot needs to follow to accomplish the task is the same as the demonstrated one, any augmentation trajectory that converges to it should be able to achieve the task (as long as it does not disturb the environment).

These augmented trajectories will contain observations that the robot may encounter when the object's pose changes, allowing the method to generalize. Alternatively, if the robot had perfect object pose estimation (As in the evaluation “Pose Estimation + Demo Replay”), it could go back to the original demonstration (relative to the object) and follow it. But this may also not be possible as the robot may hit/collide with the environment. Is this correct? If so, this information can be further highlighted in motivation or something like problem formulation.

Regarding resetting, I have one question. In “Validity Condition for Augmentation Trajectories,” condition 2 says that if the environment is disturbed, corresponding augmentation trajectories will not be recorded. However, if the environment is disturbed, this may also cause the original demonstration to be invalid (e.g., the robot accidentally changing the object's position in the scene). Wouldn’t this require a reset? Wouldn't this be a limitation? If not, maybe the authors can clarify this.

**Quality Of The Limitations Section:**

3

**Questions For Rebuttal:**

I don’t have any questions for rebuttal, but the authors may clarify my questions from my review.

**Robotics Focus:**

4

**Summary Of Paper:**

This paper proposes a data augmentation technique where a robot learns how to get back to a state in a demonstrated trajectory. This allows the robot to perform the task even with a single demonstration, as the robot can follow the rest of the demonstrated trajectory after it gets back to it.

**Summary Of Recommendation:**

Authors propose a simple but effective idea for data-efficient imitation learning. Paper is well written and method is well demonstrated.

---

### Official Review · Reviewer_zWvW · 2024-07-22
**Interesting paper, but raises some questions**

**Originality:** 4
**Technical Quality:** 3
**Clarity Of Presentation:** 3
**Potential Impact:** 3
**Recommendation:** 3
**Confidence:** 4

**Review:**

(++) The authors take on an important and well-motivated problem in robot learning, ie, how to better perform imitation learning given the practical costs incurred by the paradigm (eg, needing to provide myriad demonstrations and needing to babysit the robot during data collection).



(++) The proposed approach’s reported ability to successfully imitate while relying mainly on self-supervised data collection is attractive.



(++) The authors perform an impressive set of on-robot manipulation experiments in order to demonstrate the efficacy of their algorithm, including a convincing set of ablation experiments for MILES itself.



(--) Even after reading the supplemental material, I find myself questioning the validity of the proposed approach when it comes to the so-called “environment disturbances.” As I understand it, the proposed data collection process will stop when such a disturbance is detected, where these disturbances are when, eg, an object falls over due to suboptimal robot behavior during the self-supervised data collection phase. This seems counter-intuitive since it seems that states in which a disturbance can occur would be some of the more challenging parts of the task, and so it’s not clear how we can hope that MILES would learn to imitate in tasks that run the risk of disturbance in such cases.



(--) I am concerned about the fairness of the experiments given that MILES would appear to have a pretty strong advantage in terms of its ability to reset itself to the demonstration states. What enables the approach to achieve these states? That is, _how_ is the robot moved to a random state near a demonstration waypoint? Apologies if I missed an explanation in the paper, but given that I didn’t find one, I’m assuming some kind of auxilliary controller is used to do this, and I find myself wondering if any of the comparison methods are able to leverage this kind of policy and, if not, whether or not that presents an issue in terms of fair comparison.



(--) While the method is presented as one for imitation learning in general, I can’t help but wonder if the method is really applicable to other settings beyond the manipulation domain studied here. In particular, I am skeptical that MILES would be practical for navigation tasks, where it seems very challenging to be able to place the robot at a random state close to a demonstration waypoint without first having some kind of policy capable of navigation in the first place.



(-) The proposed method seems similar—at least at a high level—to FAIL [Sun et al., ICML 2019]. It would have been nice to have seen this somehow addressed in the paper.


**Post-response update**: Thanks to the authors for the in-depth response to my review here. The responses to address a number of my concerns, and I understand that the authors will add several new discussion points to the paper in light of the questions I raised. As such, I'm happy to bump up my score.

**Quality Of The Limitations Section:**

3

**Questions For Rebuttal:**

(1) Can the authors provide any intuition as to how MILES can succeed in imitation for tasks in which environment disturbance is possible at points throughout the demonstration trajectory?



(2) Can the authors comment on how MILES is able to guide the robot to a random location near a particular waypoint and how this ability does not cause any issues of fairness when it comes to comparing to the other approaches presented?



(3) How does the propose approach compare to FAIL?

**Robotics Focus:**

4

**Summary Of Paper:**

The authors propose MILES, which they explain to be an algorithm for performing imitation from a single demonstration without requiring manual environment resets. The crux of MILES appears to be the self-supervised data collection process, which generates a large number of augmented demonstration trajectories. The authors perform an impressive set of on-robot experiments showing that MILES performs well for robot manipulation tasks.

**Summary Of Recommendation:**

While there’s a lot to like about this paper (well-motivated, novel, and fairly impressive expeirmental results), I have some important reservations that I think need to be addressed before the paper could be accepted.

---

### Author Rebuttal · Authors · 2024-08-14

We would like to thank all the reviewers and the Area Chair for their time in reviewing our paper. We have made several updates to the paper and the supplementary material to incorporate the feedback and conversations we had with the reviewers during the rebuttal phase. In the zip file we have included the updated paper and appendix with the changes marked in blue.

Please note that the paper included in the rebuttal is currently slightly longer than the maximum page length. After the final discussion phase, we will make sure to keep it inside the maximum page limit, while incorporating all comments from the rebuttal phase and any new comments that might arise from the AC-reviewers discussion phase.

---

### Decision · Program_Chairs · 2024-09-04

**Decision:**

Accept

**Comment:**

The paper propose a new algorithm for imitation learning, MILES, that requires a single demonstration and a single reset. MILES designs a self-supervised data collection process, that generates a large number of augmented demonstration trajectories. Reviews appreciate the simplicity of the method and real-world testing that includes many ablations. Reviews have also noted many shortcomings: the strong assumptions made by the proposed method, the relationships to some past works, and some counterintuitive trends in the experiments.

Authors provided a response that was considered by `dHST`, `3nym`, `onvu`, who all appreciated the response and converged on accept leaning ratings. `zWvW` gave an original rating of weak reject and didn't look through the author response or participate in any discussion. The AC also considered the paper, the reviews, and the author response.

The AC believes that the proposed method makes strong assumptions and would only work on tasks of a specific form. It seems that MILES is limited to free space motion to a precise pose (e.g. getting the key to the keyhole) and execution of a fixed motion (e.g. turning the key). MILES excels at getting to this precise pose. However, many tasks commonly considered in robot learning literature (e.g. pick an object and place it at a target location, or pick a cup and pour its content into another cup) require careful execution even after the first contact is made with the environment. Other tasks such as tossing a ball require carefully following the precise trajectory to impart sufficient momentum to the ball and not just reaching a precise pose. MILES methodology exploits this specific form of the task and the paper doesn't demonstrate if it will work for more general tasks. The paper's introduction should prominently note these limitations.

Furthermore, even for tasks of this form, the methodology is not robust to accidental changes in the environment, e.g. the screw driver changing pose in the gripper. Author response argues that this can be detected and data collection can be terminated. If this were to happen early on in data collection, then MILES performance wouldn't be much better than just training on the 1 demonstration.

On the experimentation front,
- Generalization to distractor objects in experiments presented in the paper likely comes from the use of wrist camera as the distractor objects are not even visible in the wrist camera image. These claims should be appropriately toned down.
- Pose Estimation + Demo Replay already does quite comparably with MILES only outperforming it on 4/7 tasks. Authors note that Pose Estimation + Demo Replay fails because one can't precisely recover the pose. However, the MILES method itself relies on accurate estimation of whether the environment has changed or not using DINO ViT features, and authors note that it is actually quite sensitive to even slight changes in the environment. If this is so, wouldn't a "pose estimation + local search in a small neighborhood to maximize DINO ViT similarity + demo replay" already be quite strong? Furthermore, this process could be done for multiple way points along the way as opposed to just the first way point.